# Silencing of HMGA2 by siRNA Loaded Methotrexate Functionalized Polyamidoamine Dendrimer for Human Breast Cancer Cell Therapy

**DOI:** 10.3390/genes12071102

**Published:** 2021-07-20

**Authors:** Fereydoon Abedi Gaballu, William Chi-Shing Cho, Gholamreza Dehghan, Amir Zarebkohan, Behzad Baradaran, Behzad Mansoori, Soheil Abbaspour-Ravasjani, Ali Mohammadi, Nader Sheibani, Ayuob Aghanejad, Jafar Ezzati Nazhad Dolatabadi

**Affiliations:** 1Immunology Research Center, Tabriz University of Medical Sciences, Tabriz 5166-15731, Iran; fereydoon_abedi94@ms.tabrizu.ac.ir (F.A.G.); baradaranb@tbzmed.ac.ir (B.B.); bmansoori@health.sdu.dk (B.M.); 2Department of Biology, Faculty of Natural Sciences, University of Tabriz, Tabriz 51666-16471, Iran; 3Department of Clinical Oncology, Queen Elizabeth Hospital, Hong Kong, China; chocs@ha.org.hk; 4Department of Medical Nanotechnology, Faculty of Advanced Medical Sciences, Tabriz University of Medical Sciences, Tabriz 5166-15731, Iran; zarebkohana@tbzmed.ac.ir; 5Department of Cancer and Inflammation Research, Institute for Molecular Medicine, University of Southern Denmark, 5230 Odense, Denmark; amohammadi@health.sdu.dk; 6Drug Applied Research Center, Tabriz University of Medical Sciences, Tabriz 5166-15731, Iran; Abbaspour.s@tbzmed.ac.ir; 7McPherson Eye Research Institute, University of Wisconsin School of Medicine and Public Health, Madison, WI 53726, USA; nsheibanikar@wisc.edu; 8Department of Ophthalmology and Visual Sciences, University of Wisconsin School of Medicine and Public Health, Madison, WI 53726, USA; 9Research Center for Pharmaceutical Nanotechnology, Tabriz University of Medical Sciences, Tabriz 51666-16471, Iran; aghanejaday@tbzmed.ac.ir

**Keywords:** apoptosis, dendrimer, HMGA2 *siRNA*, methotrexate, breast cancer

## Abstract

The transcription factor high mobility group protein A2 (HMGA2) plays an important role in the pathogenesis of some cancers including breast cancer. Polyamidoamine dendrimer generation 4 is a kind of highly branched polymeric nanoparticle with surface charge and highest density peripheral groups that allow ligands or therapeutic agents to attach it, thereby facilitating target delivery. Here, methotrexate (MTX)- modified polyamidoamine dendrimer generation 4 (G4) (G4/MTX) was generated to deliver specific small interface RNA (siRNA) for suppressing HMGA2 expression and the consequent effects on folate receptor (FR) expressing human breast cancer cell lines (MCF-7, MDA-MB-231). We observed that HMGA2 siRNA was electrostatically adsorbed on the surface of the G4/MTX nanocarrier for constructing a G4/MTX-siRNA nano-complex which was verified by changing the final particle size and zeta potential. The release of MTX and siRNA from synthesized nanocomplexes was found in a time- and pH-dependent manner. We know that MTX targets FR. Interestingly, G4/MTX-siRNA demonstrates significant cellular internalization and gene silencing efficacy when compared to the control. Besides, the 3-(4,5-dimethylthiazol-2-yl)-2,5-diphenyl tetrazolium bromide (MTT) assay demonstrated selective cell cytotoxicity depending on the FR expressing in a dose-dependent manner. The gene silencing and protein downregulation of HMGA2 by G4/MTX-siRNA was observed and could significantly induce cell apoptosis in MCF-7 and MDA-MB-231 cancer cells compared to the control group. Based on the findings, we suggest that the newly developed G4/MTX-siRNA nano-complex may be a promising strategy to increase apoptosis induction through HMGA2 suppression as a therapeutic target in human breast cancer.

## 1. Introduction

Cancer is the second most common cause of death after cardiovascular disease [1]. Surgery, chemotherapy, radiotherapy, targeted therapy and their combination are employed to manage various types of cancers in clinics [2]. Among them, chemotherapy is the most common option to treat cancers. Following the rapid progression of nanotechnology, drugs incorporating nanoparticles have been examined to prolong blood circulation time, increase therapeutic tumor accumulation, promote anti-cancer activity, reduce undesirable side effects, and overcome multidrug resistance [3,4]. Several drug delivery systems with multiple mechanistic functions have been reported [5]. For example, polymers, dendrimers, and lipid-based delivery systems have demonstrated encouraging therapeutic activities in both in vitro and in vivo conditions [6]. 

Methotrexate (MTX) has exhibited anti-cancer activity and has a similar structure to folic acid (FA) with the ability to inhibit dihydrofolate reductase (DHFR), which ultimately impedes the synthesis of deoxyribonucleic acid (DNA), ribonucleic acid (RNA) and protein [7]. Based on the MTX affinity to the folate receptor (FR), it has been used as a ligand moiety for modifying delivery systems [8]. It has been reported that folate-modified nanoparticles employ clathrin/caveolae independent endocytosis pathway for cellular internalizations. Interestingly, this pathway is a non-destructive route that allows the cargoes to be released into the cytoplasm [9]. On the other hand, MTX single therapy at high concentration is usually associated with low efficacy and drug resistance. Thusly, MTX needs to be incorporated with other therapeutic agents such as RNA interface which have apparent advantages not only in reducing side effects and enhancing anti-cancer activity, but also overcoming drug resistance [10,11].

Small interfering RNA (siRNA) is emerging as a key strategy for the treatment of different cancers. It is able to suppress the production of oncogenic regulators involved with tumor growth and metastasis through a specific gene silencing effect [12,13]. The therapeutic activity of siRNA can exert by its interacting with the RNA-induced silencing complex (RICS) and enhances cleavage of complementary sequence on messenger RNA (mRNA) [14,15]. High mobility group protein A2 (HMGA2) is a family of nuclear proteins consisting of an N-terminal tail that is able to recognize and bind the AT-rich sequence in the minor groove of DNA, and a C-terminal part that may regulate their interplay with DNA and proteins [16]. This protein is a non-histone transcription factor and can regulate the transcription of multiple genes via direct attachment to AT-rich part of DNA and changes the chromatin structure [17]. In addition, HMGA2 influences various biological behavior, including cell cycle regulation, DNA damage repair, senescence, and apoptosis processes. Furthermore, HMGA2 is able to inhibit the induction of tumor cell apoptosis by the protection of telomerase [18] and has been noted as one of the most important transcription factors that have a key role in cell development along embryogenesis. Additionally, according to recent studies, the overexpression of HMGA2 has been discovered in many types of cancers and is involved in the initiation, development, and metastasis of cancer cells [19,20]. Furthermore, HMGA2 is implicated in regulating key human genes that are linked to mesenchymal cell lineage differentiation, adipogenesis, and humans—embryonic stem cells (ESCs). Moreover, the accumulating evidence suggests that HMGA2 protects tumor cells from apoptosis via activating phosphoinositide-3-kinase (PI3K/Akt) pathway and increasing Bcl2 expression to exert an anti-apoptotic effect. Previously, we showed that down regulation of HMGA2 expression could induce apoptosis in various types of cancer cells [21,22,23]. However, due to existing high nucleases in the serum and endolysosome, it is hard for siRNA alone to meet the target site or cells cytoplasm [24]. Therefore, it seems that the development of non-viral delivery system for transfecting nucleic acid segments into the cancer cells with enhanced efficacy and safety is indisputable.

Poly-amidoamine dendrimers are a kind of biocompatible, nonimmunogenic, highly branched, polymeric nanoparticle, with a spherical structure and uniform size [25]. More interestingly, they exhibit high terminal functional groups and abandon buffer capacity that facilitates endosomal rupture and release into the cell’s cytoplasm, consequently making them an efficient vehicle for drug and gene delivery [26]. Several studies have been employed poly-amidoamine dendrimers as a gene delivery nanosystem [1,27,28]. However, some drawbacks, including cytotoxicity and poor targeting, limit their application. Fortunately, the unique properties of the poly-amidoamine dendrimers surface enable them to modify with a controlled number of targeting ligands that can reduce cytotoxicity, enhance transfection efficiency, and increase targeting ability [24]. In the current study, we first synthesized G4/MTX as a gene delivery nanosystem, then adjusted it to specific HMGA2 siRNA for fabricating a G4/MTX-siRNA nanocomplex. Meanwhile, cellular uptake, in vitro HMGA2 gene silencing efficiency, selective cytotoxicity, and cell apoptosis of G4/MTX-siRNA were examined on MCF-7 and MDA-MB-231 (human breast cancer) cells. As expected, our prepared nanocomplex selectively accumulated in the cytoplasm of FR expressing cancer cells and indicated high HMGA2 gene silencing efficiency, which finally improved the induction of cell apoptosis. 

## 2. Materials and Methods

### 2.1. Materials

Ethylenediamine core polyamidoamine generation 4, dimethyl sulfoxide (DMSO), 2-(4-amidinophenyl)-6 indolecarbamidine dihyformaldehyde (DAPI), fluorescein isothiocyanate (FITC), and methotrexate were purchased from Sigma Aldrich (St. Louis, MO, USA). Targeting carboxyfluorescein (FAM) tagged HMGA2 siRNA (sense: 5′- GCACUUUCAAUCUCAAUCUtt-3′, anti- sense: 5′-AGAUUGAGAUUGAAAGUGCtt-3′) was prepared from Santa Cruz. Phosphate-buffered saline (PBS), Roswell Park Memorial Institute 1640 (RPMI), penicillin-streptomycin (10,000 U/mL), trypsin-EDTA (0.25%), fetal bovine serum (FBS), Annexin V-FITC, trimethylamine (TEA), and 3-(4,5-dimethylthiazol-2-yl)-2,5-diphenyl tetrazolium bromide (MTT) were obtained from Gibco. 

### 2.2. Synthesis of G4/MTX Nanosystem

Based on the cellular uptake controlling test, 1 G4 dendrimer to 16 MTX (1:16) was an efficient and optimized ratio to synthesis G4/MTX nanosystem. For this purpose, 77 mg MTX with MW = 454.44 g/mol was separately dissolved in PBS (pH 7.4) and mixed with 1 mL of G4 dendrimer 10% *w/v* (100 mg, MW = 14,215 g/mol) in a mixture of 5 mL of distilled water and slightly vortexed for 2 h. Afterward, the prepared solution containing G4/MTX nanovehicle evaporated to eliminate methanol from the environment. In the next step, G4/MTX was dialyzed to eliminate unreacted MTX against DI water using a dialysis batch with a molecular weight cutoff (MWCO) of 3.5 kDa for 24 h. Finally, the obtained G4/MTX nanoparticle was lyophilized and used for in vitro experimental assays. 

### 2.3. Preparation of G4/MTX-siRNA Nanocomplex 

An adequate amount of the lyophilized G4/MTX sample was dissolved in RPMI (pH 7.4). Then, G4/MTX-siRNA nanocomplex was prepared via incorporating siRNA solution (100 µM) with a specified amount of G4/MTX solution, followed by incubation at room temperature for 20 min, and gently vortexed. Finally, the prepared nanocomplex was subjected to the characterization process. 

### 2.4. Characterization f G4/MTX Nanosystem and G4/MTX-siRNA Nanocomplex

To observe the probable interaction of MTX with G4 dendrimer for preparing G4/MT nanosystem, UV-Vis spectroscopy, Fourier-transform infrared (FTIR), and nuclear magnetic resonance (NMR) spectroscopies were applied. The hydrodynamic particle size of G4 dendrimer, G4/MTX, G4-siRNA, and G4/MTX-siRNA nanocomplex was measured by dynamic light scattering (DLS) and zeta plus analyzer (Zeta-sizer, Malvern nano ZS, Worcestershire, UK). Moreover, the real size and morphology of G4/MTX in complexing with HMGA2 siRNA was visualized by atomic force microscopy (AFM) (Microsynth, Balgach, Switzerland).

### 2.5. Gel Retardation Assay Electrophoresis

The G4/MTX-siRNA nanocomplex at various N/P ratios (1:1, 5:1 10:1, and 20:1 NH2/Phosphate (N/P) ratio G4/MTX to siRNA) was synthesized and their mobility and stability were assessed by using agarose gel-retardation assay. This aim was carried out by loading 20 µL each of solutions (free siRNA and nanocomplex) into wells of 2% agarose gel (RNase free Tris-acetate/EDTA buffer) containing DNA safe stain for 40 min under 80 V. Ultimately, any retardation on the gel electrophoresis was visualized using gel doc to confirm the successful formation and stability of the G4/MTX-siRNA nanocomplex.

### 2.6. MTX and siRNA Release from the G4/MTX-siRNA Nanocomplex

The release of MTX and siRNA from the G4/MTX-siRNA nanocomplex was separately considered and measured by dialysis method. Briefly, 1 mg of G4/MTX-siRNA was dissolved in 2 mL of PBS and then placed into the dialysis bag with MWCO 3.5 kDa and 12 kDa for MTX and siRNA, respectively, and immersed in 50 mL of PBS (10 mM, pH 7.4 and 5.5). The release system was kept at 37 °C and shaken at 160 rpm. At predetermined time intervals, 1 mL of each sample was withdrawn and replaced with fresh buffer solution. The accumulative release of MTX was measured for 4 days using UV–Vis spectrophotometry at 310 nm. In addition, the amount of released FAM-siRNA was monitored in fluorescence intensities of FAM-siRNA excitation/emission at 488/520 nm by A Jasco FP-750 spectrofluorimeter (Kyoto, Japan). 

### 2.7. Cell Culture and In Vitro Transfection Experiment 

MCF-7 and MDA-MB231 cancer cell lines were obtained from the national cells bank of Iran (Pasteur Institute, Tehran, Iran) and grown in RPMI-1640 containing 10% FBS, 2 mM L-glutamine, 100 unit/mL penicillin, and 100 µg/mL streptomycin in a humidified incubator with a temperature of 37 °C and an atmosphere containing 5% CO_2_. 

### 2.8. Cellular Uptake Study

MCF-7 and MDA-MB-231 cells were cultured in 12 wells at a density of 20,000 cells/well and allowed to attach and grow for 24 h. Before treatment, the medium was thoroughly eliminated and the cells were rinsed with PBS once. Then, the cells were incubated with different particles including G4/FITC (40 µg/mL dissolved in serum-free RPMI medium), G4/MTX (1:16)/FITC (40 µg/mL dissolved in serum-free RPMI medium), G4/MTX (1:32)/FITC (40 µg/mL dissolved in serum-free RPMI medium), and Fam-labelled HMGA2 siRNA complexed with dendrimers (40 µg of G4/200 nmol of siRNA and 40 µg of G4/MTX (1:16)/200 nmol of siRNA dissolved in 1000 µL of serum-free RPMI medium) in the absence and presence of 1 mM folic acid (FA) for 4 h. At the end of the incubation time, the media was removed and thoroughly washed with cold PBS three times and then the cells were trypsinized and collected through centrifugation at 1000 rpm for 10 min. Finally, the collected cells were subjected to flow cytometry analysis using Flow cytometer equipment (MACS Quant 10, Miltenyi Biotech GmbH, Gladbach, Germany). The obtained data were analyzed by FlowJo software (Treestar, Inc., San Carlos, CA, USA). The main fluorescence intensity (MFI) values of FITC for quantifying G4 dendrimer-based nanocarriers and FAM for quantifying HMGA2 siRNA in the cells were studied.

Cellular uptake was also monitored via fluorescence microscopy imaging. The cells were grown and seeded onto the glass coverslips placed in a six-well plate at density 50,000 cells/well and allowed them to reach about 70% confluency, then exposed to the same treatment groups as those in flow cytometry analysis assay above. At the end of incubation times, the cells were washed with cold PBS and fixed with 4% formaldehyde for 20 min, washed with PBS, and adjusted to the solutions containing 0.1% Triton X-100 for 5 min, then washed again with PBS three times. Furthermore, the nuclei of the studied cells were stained with DAPI for 5 min, washed with cold PBS three times, and finally visualized by fluorescence microscopy (Olympus Microscope Bh2-RFCA, Tokyo, Japan). 

### 2.9. Gene Expression

The studied cells seeded into the 6-well plates with 200,000 cells and treated with G4/MTX, G4-siRNA, and G4/MTX-siRNA as described earlier for 48 h. Total RNA was isolated using RiboEx reagent (GeneAll Biotechnology, Seoul, Korea) according to the manufacturer’s protocol. The cell was harvested and washed with PBS. Then 1 mL of RiboEx reagent added to the cell pellet and mix gently, and then 250-µL chloroform was added and centrifuged at 12,000 rpm for 20 min in 4 °C. The aqueous phase transfer to another tube and isopropanol was added. Then, the solution was incubated for 30 min in −20 °C. SSubsequently, the solution was centrifuged at 12,000 rpm for 20 min in 4 °C. After that, the pellet was washed with 70% ethanol and the RNA pellet suspend on DEPC. The concentration of the isolated RNA was measured by a NanoDrop spectrophotometer (Thermo Fisher Scientific, Waltham, MA, USA). RNA inverted to cDNA by the RT-PCR method according to BioFact protocol. Briefly, one microgram of isolated RNA, 10 µL of premix enzyme contain RT enzyme, DNATP and RT buffer, 0.5 µL oligo dT, and 0.5 random hexamers were added to a microtube and added DEPC up to 20 µL. The reaction was transferred to a thermocycler instrument (BioRad, Hercules, CA, USA) with a thermal protocol that included 30 min at 50 °C and 5 min at 95 °C. Quantitative real-time PCR was carried out in the light cycler system (Roche, Mannheim, Germany) using the SYBR green method. Briefly, 5 µL SYBR green master mix (BioFact, Daejeon, Korea), 0.5 µL of 4 pmol specific primers (Table 1), 4 µL H_2_O_2_, and 0.5 µL cDNA was added to qRT-PCR strips. The thermal protocol for qRT-PCR analysis was 800 s preincubation at 95 °C for enzyme activation, followed by 10 s at 95 °C, 35 s at 60 °C and 20 s at 72 °C. β-actin was used as an internal control. Relative *HMGA2* mRNA expression was measured with the 2^−ΔΔCt^ method.

### 2.10. Western Blot Analysis

The total protein was isolated from MCF-7 and MDA-MB231 cells treated with G4/MTX, G4-siRNA, and G4/MTX-siRNA nanocomplex by radioimmunoprecipitation assay (RIPA) protein extraction kit (Santa Crus Biotechnology, Santa Cruz, CA, USA). The protein concentration was measured by NanoDrop spectroscopy. The sodium dodecyl sulphate–polyacrylamide gel electrophoresis (SDS-PAGE) with 4% stacking and 12% running gel was performed to separate HMGA2 protein based on their molecular weight. Fifty milligrams of each extracted protein loaded into each well of SDA-PAGE. Then, the protein blots to polyvinylidene fluoride (PVDF) membrane (Roche, Mannheim, Germany) using the semidry immunoblotting system (BioRad). The membrane was blocked with 0.5% Tween 20 in PBS for 2 h in RT condition, then incubated with a monoclonal antibody against HMGA2, and β-actin as a reference protein overnight at 4 °C (1:1000; Santa Cruz Biotechnology, Dallas, TX, USA). Then, the membrane was incubated with a secondary rabbit anti-goat conjugated to HRP (1:5000; diluted in PBS) for 1 h at RT condition. The HMGA2 and β-actin specific bonds were visualized using the electro-chemiluminescence method (Roche) and the bonds were imaged by Western blot imaging system (Sabz Co., Urmia, Iran). 

### 2.11. Cell Viability Assay

Cell viability was studied by MTT assay. For this aim, 15,000 cancer cells (MCF-7, MDA-MB231) were distributed in 96-well plates at 200 µL of culture media and incubated to reach more than 70% cell confluency. After 24 h incubation, the cells were incubated with growth media free of FBS (200 µL) containing G4, MTX, G4/MTX, G4-siRNA, and G4/MTX-siRNA nanocomplex at the G4 amount ranging from 5 to 80 µg/mL. After 4 h incubation, the media were replenished with 200 µL of fresh growth media containing 10% FBS and incubated for 48 h. Next, the supernatant in each well was eliminated and replaced with 200 µL of fresh media containing MTT solution (2 mg/mL), then additionally incubated for 4 h. The incubated medium was changed with 200 µL/mL of DMSO and after 30 min of incubation. The absorption of the plates was read at a wavelength of 570 nm by ELISA reader (Sunrise; Tecan Co., Grödingen, Austria).

### 2.12. Apoptosis Assay

#### 2.12.1. Annexin V/Propidium Iodide Staining

MCF-7 and MDA-MB231 cells at the density of 200,000 cells/well in six-well plates were cultivated and incubated in 2 mL RPMI containing 10% FBS, until reach over 70% confluency. Following 24 h incubation, the cells were treated with MTX, G4/MTX, G4-siRNA, and G4/MTX-siRNA in 2 mL FBS-free RPMI for 4 h. Then, the medium was replenished with 2 mL fresh media containing 10% FBS and incubated again for 48 h. At the end of the incubation time, the cells were harvested and detached by adding diluted trypsin in each well, washed twice in PBS, and re-suspended in binding buffer at ice-cold temperature. Finally, the cells were detached and stained with Annexin V/PI kit to measure the rate of cell apoptosis. Flow cytometer equipment was used for analyzing cell apoptosis. The obtained data were analyzed by using the FlowJo software package (Treestar, Inc., San Carlos, CA, USA). 

#### 2.12.2. DAPI Staining

The cancer cells (200,000 per well) were seeded and cultivated onto the glass coverslips placed in a six-well plate. The cells with sufficient confluency (70%) were incubated with MTX, G4/MTX, G4-siRNA, and G4/MTX-siRNA in a growth media free of FBS. Following 4 h incubation, the media was changed with 2 mL fresh media containing 10% FBS and additionally incubated for 48 h. Then, the cells were washed with PBS three times and stained with DAPI as described earlier.

### 2.13. Statistical Analysis 

The statistical analysis of the obtained data was carried out using GraphPad Prism software, version 4.0 (GraphPad, San Diego, CA, USA). The results are expressed as means ± standard deviation (SD) and n indicating the numbers of biologically and technically repeats. The Student’s *t*-test and two-way analysis of variance (ANOVA) were used to compare the groups with parametric data. *p* < 0.05 was considered to illustrate a statistically significant difference.

## 3. Results

### 3.1. Confirmation of G4/MTX Nanosystem Synthesis

MTX as both a ligand and therapeutic moiety was combined with G4 dendrimer nanoparticles to synthesis G4/MTX nanosystems. The possible interaction between G4 dendrimer and MTX was determined by FTIR spectroscopy. Based on the FTIR obtained result, G4/MTX showed lower width absorption peaks at 3200–3400 cm^−1^ than free G4 dendrimer and MTX. A bond at 1645.13 cm^−1^, 3447.19 cm^−1^, and 1555.67 cm^−1^ are appeared and attributed to C=C, primary amines (N-H stretching) and N-H of amid II bonds respectively that is assigned presence both MTX and G4 dendrimer (Figure 1a). The structural characterization of G4 dendrimer in D_2_O and G4/MTX in DMSO was confirmed by H-NMR for detecting functional groups (Appendix A). As shown in Figure 1b, the peaks in 2.2–3.4 ppm are observed for G4/MTX NMR spectra that belong to G4 dendrimer methylene protons. In addition, peaks belonging to the aromatic region of MTX were observed in 5.6 ppm, 6.6 ppm, and 8.5 ppm that correlate to the aromatic ring of MTX. In this section, we have to mention that, because of the ionic interaction between the negative charge of the MTX acidic group and the positive charge of NH_2_ group on the dendrimer surface, we could not show a specific bond peak in the FTIR and NMR. However, by using these two techniques, we tried to show only the shift of IR and NMR peaks (interactions) in G4, MTX and G4/MTX. It is noteworthy that we characterized this interaction by another complement experiment in the next steps.

### 3.2. Characterization of G4/MTX Nanosystem and G4/MTX-siRNA Nanocomplex

For further study, we measured the hydrodynamic particle size of G4 dendrimer and its derivatives using DLS, and the results are represented in Figure 1c. It was shown that the mean size of G4 dendrimer nanoparticles increased from 6 ± 1.7 nm to 12 ± 2.6 nm in G4/MTX. Interestingly, combining of MTX with G4 dendrimer, zeta potential was decreased from 20 ± 3 mV in bare G4 nanoparticles to 11.3 ± 3.6 mV in G4/MTX, indicating that the number of NH_2_ on G4 dendrimer was reduced through covering with MTX. The efficacy of attached MTX in G4/MTX was calculated to be 90% that measured by UV spectrophotometry after evaluation of nanoparticles by calibration curve (data was not shown). Additionally, as seen in Figure 1c, the G4/MTX-siRNA nanocomplex showed the narrow particle size of 160 ± 46.7 nm (PDI = 0.29) and suitable zeta potential—7.6 ± 3.6 mV) at N/P 20, which could be suitable for endocytosis-mediated cellular internalizations and also represent more logical properties for drug delivery. It is noteworthy to mention that the above-reported size is the hydrodynamic diameter of G4/MTX-siRNA nanocomplex, so for real visualizing and size measurement we used AFM (Figure 1d). The AFM results demonstrated spherical shapes with a particle size of approximately 50 nm. 

### 3.3. HMGA2 Specific siRNA Electrostatically Loaded on the G4/MTX Nanosystem

The G4/MTX-siRNA nanocomplex was prepared via incubating of positively charged G4/MTX with specific HMGA2 siRNA. The positive nature of the G4/MTX allows siRNA to bind on the G4/MTX surface via electrostatic interactions. Agarose gel electrophoresis was used to determine the loading of HMGA2 siRNA on the G4/MTX at different N/P ratios. The numbers including 1–8 in the image of the gel retardation assay (Figure 1e) correspond to DNA ladder, siRNA alone, N/P 1:1, N/P 2:1, N/P 5:1, N/P 10:1, N/P 20:1, and N/P 40:1 respectively. The obtained result indicates, at the N/P 20:1, siRNA is completely loaded on the G4/MTX nanosystem that is evidenced by the elimination of siRNA bands in gel agaroses electrophoresis. 

### 3.4. MTX and HMGA2 siRNA Release Profile 

The dialysis approach was used for the determination of in vitro MTX and siRNA release from G4/MTX-siRNA nanocomplex, in a standard release medium (PBS), with pH 7.4 and 5.5 within 96 h. As seen from Figure 1f, the release of MTX from G4/MTX-siRNA was 52.3% at pH 5.5 during 48 h, whereas at the same time release was decreased to 32% at physiological pH (7.4), which related to the ionic nature of attached MTX to G4 dendrimer. Moreover, siRNA was greatly released from G4/MTX-siRNA nanocomplex (Figure 1f) at pH 5.5 (12%) than pH 7.4 during 6 h. Additionally, the same as MTX, siRNA release was higher at acidic conditions than physiological pH, so 50% of siRNA was released in approximately 29 and 48 h at pH 5.5 and pH 7.4, respectively. 

### 3.5. G4/MTX Nanosystem Efficiently Delivered HMGA2 siRNA into the FR Expressing Cancer Cells

Cellular uptake of FITC-labelled G4/MTX (1:16) and G4/MTX (1:32) nanosystems by FR-expressing cancer cells was quantified and visualized by flow cytometry and fluorescence microscopy to test and select an efficient ratio of G4 dendrimer to MTX. After incubating with two nanosystems, the cells were stained with DAPI for nuclei staining. The blue fluorescent light of DAPI (excited at 358 nm) indicates the nuclei location of cells in the samples whereas the green fluorescent light (excited at 491 nm) shows the position of FITC inside (cytoplasm) of the cells. From Figure 2a,b, it can be observed that a higher green fluorescent signal is revealed in the case of FITC-labeled G4/MTX (1:16 and 1:32) nanosystems, compared with the FITC-labeled G4 dendrimer. To confirm this result, MFI of the green channel including FITC was quantified and calculated for two ratios nanosystem. MFI values for FR expressing cells adjusted to G4/MTX (1:16) and G4/MTX (1:32) nanosystems were significantly (*p* ≤ 0.01) increased when compared with G4 dendrimer alone (Figure 2c,d). In contrast, it is difficult to observe the green fluorescent light in FR expressing cells pretreated with 1 mM of FA before incubating with FITC-labeled G4/MTX (1:16 and 1:32) nanosystem, which can further verify targeting specificity of MTX attached on the G4 dendrimer. Additionally, as compared with G4 dendrimer alone, a significant decrease in MFI value was observed for MCF-7 (*p* ≤ 0.01) and MDA-MB-231 (*p* ≤ 0.001) cells in the presence of 1 mM FA when the cells were adjusted to both nanosystems. Based on these results, we concluded that G4/MTX (1:16), or G4/MTX nanosystem, showed the best cellular uptake. Accordingly, it was selected for HMGA2 siRNA transfection study.

Aa high siRNA cellular uptake of nanoparticles is a prerequisite for efficient gen silencing. FAM-labeled HMGA2 siRNA was applied as a fluorescent probe and its green fluorescent light (excited at 488 nm) exhibits the position of delivered siRNA inside the cells. Fluorescence microscopy imaging was done to observe the intercellular localization and distribution of G4-siRNA and G4/MTX-siRNA nanocomplex. As shown in Figure 3a,b, the fluorescence was placed into the cytoplasm after cells exposed to siRNA-containing treatment groups showing that siRNA is effectively delivered into FR expressing cancer cells by G4 dendrimer and G4/MTX nanosystem. Consistent with this result, the uptake of FAM-labeled HMGA2 siRNA by the studied cells was also quantified to measure MFI using of flow cytometry. Based on the MFI values, G4/MTX-siRNA was significantly taken up by MCF-7 (*p* ≤ 0.001) and MDA-MB-231 (*p* ≤ 0.01) cells compared to the G4-siRNA treatment group. Surprisingly, there was no negligible green fluorescent light when the FR expressing cells were pretreated with 1 mM of FA before exposure to the siRNA consisting treatment group. Moreover, in the presence of FA, MFI values of both FR overexpressing cancer cells were significantly (*p* ≤ 0.001) decreased in G4/MTX-siRNA treated group than G4-siRNA. 

### 3.6. HMGA2 Gene and Protein Expression Profile

Herein, we applied the MTX functionalized G4 dendrimer nanosystem G4/MTX to deliver *HMGA2* siRNA into MCF-7 and MDA-MB-231 cancer cells, and studied the effect caused via *HMGA2* silencing on the mentioned cells. After both cell lines were transfected with G4/MTX-siRNA for 48 h, the expression of HMGA2 mRNA was investigated by real-time qPCR. It could be observed from Figure 4a that G4/MTX-siRNA nanocomplex and G4-siRNA polyplex treatment groups significantly (*p* < 0.001) decreased the *HMGA2* mRNA expression level in FR expressing cancer cells compared to the G4MTX nanosystem. The HMGA2 protein level was determined by Western blotting assay. As seen from Figure 4b,c, the delivery of *HMGA2* siRNA through G4/MTX significantly (*p* < 0.001) decreased the protein expression of HMGA2, while G4/MTX did not visibly alter the expression of HMGA2 protein level.

### 3.7. In Vitro Cytotoxicity Study

The cytotoxicity of G4, MTX, and G4/MTX to MCF-7 and MDA-MB-231 cells in the concentration range of 5–80 µg/mL were tested using MTT assay for 48 h. As seen from Figure 5a,b, the G4 dendrimer that was not loaded with MTX or siRNA illustrated fairly lower cytotoxicity even at a high concentration (80 µg/mL) in both cell lines. Moreover, free MTX as therapeutic and ligand moiety could not reduce the viability of cells and almost 80% of the cells were viable following 48 h incubation which may be explained by lower cellular uptake of MTX. Figure 5a,b shows that the cytotoxicity of the G4/MTX nanosystem was measured to determine whether it could enhance the sensitivity of FR expressing cells to MTX. G4/MTX treatment displayed more significant cell viability reduction ability for all concentrations compared to the MTX alone treatment group. To investigate the HMGA2 silencing effect on cell viability, FR expressing cancer cells were treated with different concentrations of G4/MTX-siRNA and G4-siRNA treatment groups. The result proved that G4/MTX containing specific HMGA2 siRNA nanocomplex significantly further reduced cell viability relative to G4/MTX nanosystem. For instance, HMGA2 silenced MCF-7 and MDA-MB-231 displayed 66% and 70% cell death for 80 µg/mL of G4/MTX-siRNA treatment group, while treatment with G4-siRNA at the same amount results in only 40% and 50% cell death, respectively (Figure 5a,b). 

### 3.8. Suppression of HMGA2 Enhanced Cell Apoptosis

To test HMGA2 effects on cell apoptosis, we performed annexin V-FITC/PI and DAPI stainings on HMGA2 silenced human breast cancer cells. The annexin V/PI staining cells indicated that, in the absence of HMGA2 siRNA treatment group (G4/MTX), the rate of apoptosis was 12.42% (*p* < 0.05) and 8.48% (*p* < 0.05) for MCF-7 and MDA-MB-231 compared to untreated cells respectively (Figure 6a,b). However, significant apoptosis was not observed for cells that were incubated with MTX alone. HMGA2 silenced MCF-7 and MDA-MB-231 cells showed 23.28% (*p* < 0.01), and 27.6% (*p* < 0.01) apoptosis for G4/MTX-siRNA treatment group while G4-siRNA displayed 21.6% (*p* < 0.01) and 15.94% (*p* < 0.01) apoptosis respectively in comparison with apoptosis in the control group (Figure 6a,b). In agreement with these results, DAPI staining also exhibited that HMGA2 silencing induced the highest intensity of chromatin fragmentation or nuclear condensation (as typical characteristics of cell apoptosis) in FR expressing cancer cells compared to G4/MTX and the control group (Figure 7). 

## 4. Discussion

It has been reported that HMGA2 acts as an oncogene and is overexerted by various cancers, including colon cancer, ovarian cancer, pancreatic cancer, and breast cancer [23,29]. Silencing HMGA2 expression using siRNA is expected to be an efficient approach to suppress breast cancer cell growth. However, the successful delivery of siRNA into the cytoplasm is the essential problem in using it for reducing gene expression [30]. Therefore, achieving a suitable delivery system with the potential of enhancing siRNA transfection efficiency can facilitate and exert effective gene silencing. Generation 4 PAMAM dendrimer is an efficient nano-vehicle with a peripheral positive charge and nano-size scale which allows it to be used for in-vitro and in-vivo gene transfection and gene delivery, respectively. To increase transfection efficiency and selective ability, some targeting moieties have been attached on the G4 dendrimer, including hyaluronic acid [31], folic acid [32], and cRGD peptide [25]. In the current study, we proved that transferring anti-HMGA2 siRNA using MTX functionalized G4 dendrimer (G4/MTX) nanosystem lead to reduced HMGA2 gene expression and subsequently improved apoptosis mediated cell death in FR overexpressing cancer cells. We have synthesized a targeted siRNA delivery system by simple complexing G4 dendrimer with MTX to prepare the G4/MTX nanosystem and analyzed its physiochemical characterizations. Based on the FTIR, H-NMR, and DLS obtained observations, we speculate that MTX is loaded onto the G4 dendrimer and type of attachment, mostly carried out via ionic interactions between COOH from MTX with NH_2_ of G4 dendrimer that is consistent with previous reports [33]. This result is also supported by previous reports and confirmed by a study which concluded that the acidic drugs such as MTX widely interact with dendrimer nanoparticles at physiological pH through ionic, electrostatic, hydrogen bond, and hydrophobic interactions [34]. For achieving an ideal N/P ratio for HMGA2 siRNA loading on the G4/MTX nanosystem via electrostatically interactions manner, several N/P ratios were tested using agarose gel electrophoresis. The data indicated that the G4/MTX nanosystem was not able to retard HMGA2 siRNA mobility thoroughly at lower N/P ratios, but by increasing flowing N/P up to 20, efficient retardation of HMGA2 siRNA migration was observed in the agarose gel. An increase in the main particle size and reduction in the zeta potential were revealed for bare G4 dendrimer and G4/MTX nanosystem after complexing with HMGA2 siRNA which confirm the successful attachment of siRNA on the surface of nanosystems [35]. To meet high tumor inhibition, therapeutic agents should be quickly and successfully released after internalizing into the tumor cells. Based on the obtained data, both MTX and HMGA2 siRNA release from the G4/MTX-siRNA nanocomplex was higher at pH 5.5 compared with pH 7.4, which can refer to the stability of the prepared system at physiological condition. However, the strong adsorption (ionic bonding) of MTX onto the dendrimers will be broken only at pH = 4.8–5.5 (pKa MTX). This would support therapeutic agents’ dissociation from G4 dendrimers inside the endolysosome, and consequent releasing the MTX and siRNA into the cytoplasm. 

It is reported that ligands on the surface of nanoparticles can profoundly change the behavior of functionalized nanocarriers. Such an effect was reported in some previous studies [36,37]. Several studies have shown that MTX is able to bind and target FRs [38,39]. Depending on this ability, we decided to use MTX as therapeutic and ligand moiety to modify the G4 dendrimer for targeting FR expressing cancer cells. We observed that the G4/MTX nanosystem is highly taken up by FR expressing cancer cells compared to the G4 dendrimer alone, suggesting the enhanced internalization of the G4/MTX nanosystem through FR-mediated endocytosis [40]. However, this finding was confirmed by a significant decrease in cellular uptake when the cells were preincubated with FA before being treated with G4/MTX nanosystem. In agreement with our result, the increased cellular uptake of covalently conjugated MTX to G5 dendrimer was shown previously, which demonstrates the active targeting ability of MTX [41]. siRNA transfection efficiency is relatively associated with cellular uptake. A higher cellular uptake of siRNA generally leads to impressive gene silencing, but the higher transfection efficiency could not guarantee strong gene silencing. On the other hand, after nanocarrier-siRNA successfully passed through the cell membrane, they will often meet acidic vehicles, including endosomes and lysosomes [25]. To achieve ideal gene silencing by siRNA, it should safely reach the cytoplasm. The G4/MTX-siRNA nanocomplex displayed the highest cellular uptake compared to G4-siRNA polyplex in FR expressing cancer cells, indicating that G4/MTX could successfully penetrate across cell membrane and internalize inside the cells. Moreover, these results suggest that MTX played an important role in mediating the effective cellular uptake of G4/MTX and G4/MTX-siRNA nanocomplex. It has been proved that MTX uses FR-mediated endocytosis pathway to enter the cells [40]. Besides, some studies reported that FR internalizations are mediated via clathrin-caveolae independent endocytosis [9]. Interestingly, the clathrin-caveolae independent endocytosis pathway is able to deliver the cargoes directly into the cytoplasm without meeting the lysosome route. Based on this information, we can conclude that G4/MTX is an efficient nanosystem for the safely delivery of HMGA2 siRNA. The HMGA2 as target gene and protein was selected because recent studies have demonstrated that it has apoptotic effects and is frequently overexpressed by different cancers [23]. Therefore, its silencing could conduct cancer cells towards apoptosis-mediated cell death. As expected, HMGA2 gene and protein downregulation were strongly in cells transfected with G4/MTX-siRNA nanocomplex relative to the control group showing that the gene knockdown is only occurred by HMGA2 siRNA alone. In the next part of this study, we investigated the consequence of siRNA-mediated HMGA2 gene silencing on the viability of MCF-7 and MDA-MB-231 cells. Looking at the results of cell viability assay, we find that G4/MTX containing HMGA2 siRNA is a more effective treatment to reduce cell viability than other treatment groups, which maybe a result of the synergistic effect of two therapeutic agents (MTX and HMGA2 specific siRNA) on cancer cells via efficiently exerting cell death in complementary pathways. In agreement with our study, Naghizadeh et al. concluded that, following the use of siRNA against HMGA2, cell proliferation is significantly inhibited [42]. Apoptosis is a fundamental process that occurs in multicellular organisms in order to eliminate abnormal or unsensational cells and is responsible for different biological functions [43]. It is triggered via two major pathways, including the extrinsic and intrinsic pathways [44]. Interestingly, the tumor cells with the lowest expression of HMGA2 have indicated more apoptosis than HMGA2 overexpressed cells. Indeed, targeting HMGA2 may be an effective approach for inducing cell apoptosis. Results from AnnexinV/PI and DAPI staining exhibited that G4/MTX-siRNA indicated a greater ability to induce apoptosis compared to other treatment groups. This indicated the importance of MTX (as ligand and therapeutic agent) which allows the G4/MTX-siRNA nanocomplex to recognize and bind to the FR and actively internalize into the FR expressing cells. It has been reported that HMGA2 can facilitate cell apoptosis by increasing the caspase-2 activation and permeabilization of the outer membrane of mitochondria [45]. Additionally, it was proven that suppression of the HMGA2 gene induces cell apoptosis by controlling the intrinsic apoptosis pathway [46]. In agreement with this, MTX also exerted cell apoptosis in an intrinsic apoptosis pathway [47]. Therefore, we can conclude that the co-delivery of two therapeutic agents with similar and simultaneous effect synergistically improves the rate of apoptosis. A greater apoptotic effect of G4/MTX-siRNA is also carried out synergistically. To thoroughly understand the consequent effects of our synthesized nano-siRNA formulation, more evaluations, including in vivo study and data, are needed. We specifically report the apparent treatments of this study that showed nanoscale size, controlled release profiles, selective cellular uptake, and augmented transfection efficiency, indicating the advantages of dendrimer modification with MTX in complexing with HMGA2 siRNA. However, some properties, including high cost, can be described as the main disadvantage of the nan-siRNA complex, making further progress difficult. 

## 5. Conclusions

In summary, we successfully prepared a novel type of cancer-targeting functionalized G4/MTX-siRNA nanocomplex for the efficient delivery of MTX and anti- HMGA2 siRNA to target and treat folate expressing human breast cancer (MCF-7 and MDA-MB-231) cells. Our data indicate that G4/MTX-siRNA was able to strongly internalize and accumulate in the cytoplasm of cells, resulting in significantly apoptosis-mediated cell death by specific silencing HMGA2 expression. Additionally, this outcome is more encouraging in order to use MTX as a ligand and therapeutic moiety for functionalizing non-viral gene delivery systems to target FR expressing cancer cells with potential lower off-target consequences. Overall, our findings in this research can be useful for other researchers in the design of a more effective and precise nanosystem for targeting and combating breast cancer, which presents high mortality. 

## Figures and Tables

**Figure 1 genes-12-01102-f001:**
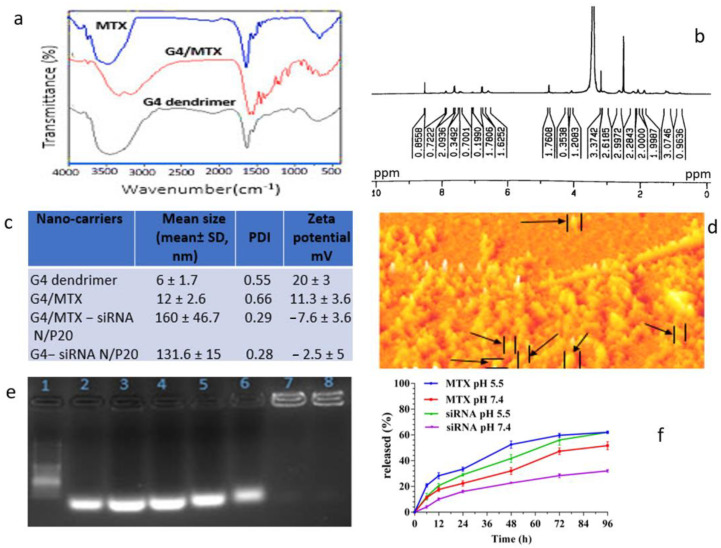
Physiochemical characterization of G4 dendrimer-based nanocarriers. (**a**) FTIR spectra of G4 dendrimer, G4/MTX and MTX; (**b**) NMR spectra of G4/MTX; (**c**) the hydrodynamic particle size, PDI and zeta potential values of dendrimer based nanosystems that were analyzed using DLS; (**d**) atomic force microscopic (AFM) image of G4/MTX-siRNA nanocomplex at N/P 20. (**e**) Representative image of gel retardation assay for G4/MTX-siRNA at different N/P ratios. (**f**) In vitro MTX and siRNA release profiles of G4/MTX-siRNA at 37 °C in PBS with pH 7.4 and pH 5.5 during 96 h period.

**Figure 2 genes-12-01102-f002:**
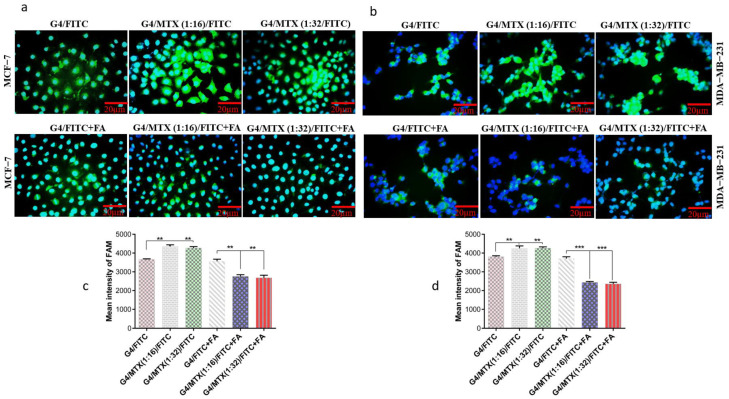
Intercellular uptake of dendrimers carriers by FR expressing cancer cells. Fluorescence microscopy images (scale bar 20 µm) of MCF-7 cells (**a**) and MDA-MB-231 cells (**b**) that incubated for 4 h with FITC-labeled G4 dendrimer, G4/MTX (1:16) and G4/MTX (1:32) in the absence and presence of FA (1 mM). Blue and green represent DAPI stained nucleus and FITC-labeled dendrimer carriers respectively. MFI quantified cellular uptake in the absence and presence FA (1 mM) for MCF-7 cells (**c**) and MDA-MB-231 cells (**d**) when cells were treated with FITC-labeled G4 dendrimer and G4/MTX (1:16 and 1:32) nanosystems. The data were expressed mean ± SD (*n* = 3). Statistic significance: ** *p* values < 0.01; *******
*p* values < 0.001.

**Figure 3 genes-12-01102-f003:**
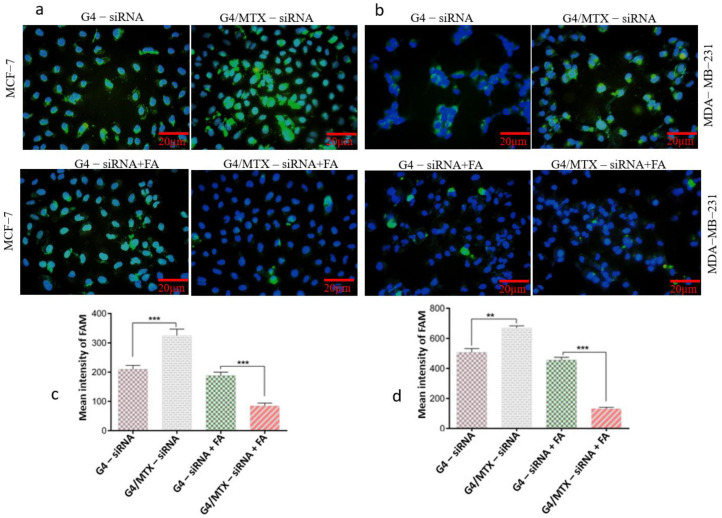
Intercellular uptake of FAM—labeled siRNA in complexing with G4 dendrimer and G4/MTX nanosystem by FR expressing cancer cells. Intercellular localization of fluorescence-labeled G4-siRNA and G4/MTX-siRNA in MCF-7 cells (**a**) and MDA-MB-231 (**b**) cells in the absence and presence FA (1 mM) for 4 h that were imaged by fluorescence microscopy (scale bar 20 µm). Blue and green represent DAPI stained nucleus and FAM—labeled siRNA respectively. MFI quantified cellular uptake in the absence and presence FA (1 mM) for MCF-7 cells (**c**) and MDA-MB-231 cells (**d**) when cells were treated with FAM—labeled siRNA in complexing with G4 dendrimer and G4/MTX nanosystems. The data were expressed mean ± SD (*n* = 3). Statistic significance: ** *p* value < 0.01; *** *p* value < 0.001.

**Figure 4 genes-12-01102-f004:**
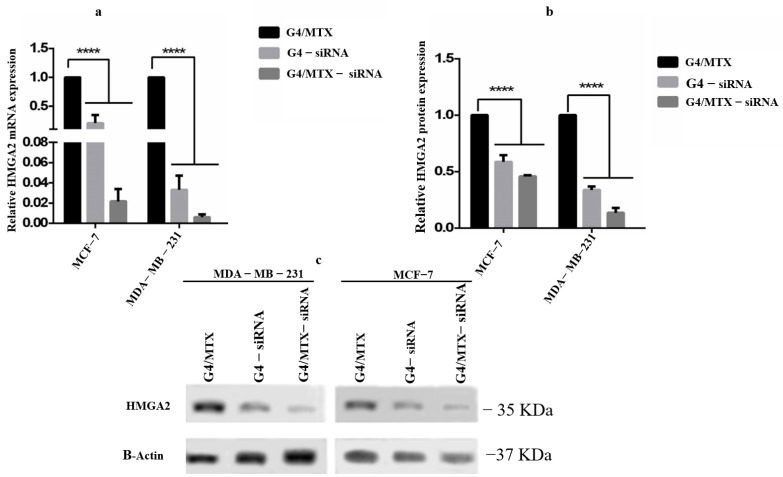
In vitro *HMGA2* gene silencing and protein suppression in FR expressing cancer cells. MCF-7 and MDA-MB-231 cells were transfected with G4-siRNA and G4/MTX-siRNA for 48 h. Relative *HMGA2* mRNA (**a**) and protein expression (**b**) were analyzed via Real time-PCR and western bolting respectively. (**c**) qualitative band of HMGA2 and β-actin proteins after siRNA was transfected. The results were expressed as mean ± SD (*n* = 3); **** *p* value < 0.0001 vs. G4/MTX nanosystem as control).

**Figure 5 genes-12-01102-f005:**
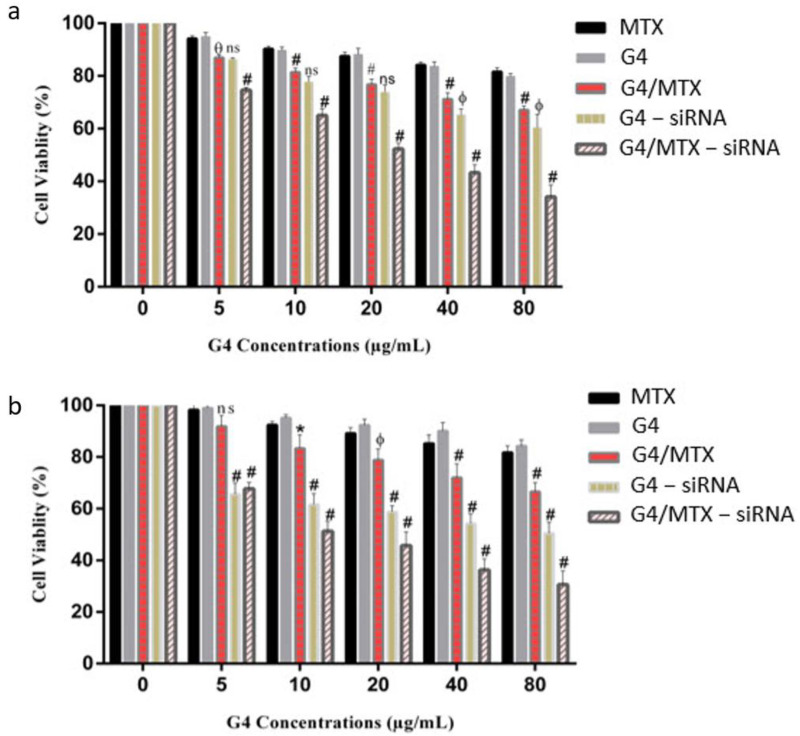
Viability of MCF-7 (**a**) and NDA-MB-231 (**b**) cancer cells after treatment with MTX, G4, G4/MTX, G4-siRNA and G4/MTX-siRNA nanocomplex (at siRNA concentrations 0–200 nM) for 48 h. Error bars displays standard deviations (SD) and the data are expressed mean ± SD (*n* = 3). (non-significant (ns), * *p* value < 0.05, ^ϕ^
*p* value < 0.005, ^ɵ^
*p* value < 0.0006 and ^#^
*p* value < 0.0001). Note: in both cells G4/MTX result was compared with MTX, while G4-siRNA and G4/MTX-siRNA were compared with G/MTX nanosystem.

**Figure 6 genes-12-01102-f006:**
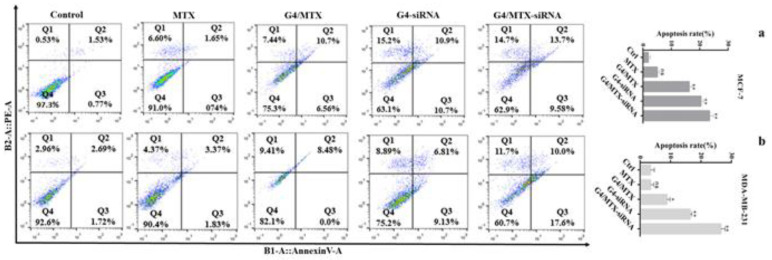
Apoptosis histogram with apoptosis percentage of MCF-7 (**a**) and MDA-MB-231 (**b**) cancer cells analyzed by flow cytometry using annexin V/PI staining after incubating with (IC25 concentration) of MTX, G4/MTX, G4-siRNA and G4/MTX-siRNA treatment groups for 48 h. G4 dendrimer amount was 40 µg/mL. Untreated cells were used as a negative control (* *p* value < 0.05 and ** *p* value < 0.01).

**Figure 7 genes-12-01102-f007:**
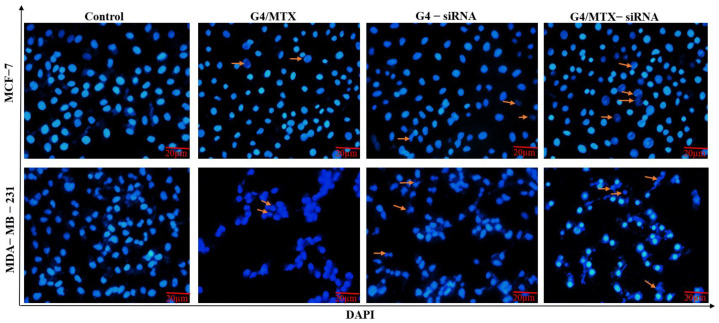
Fluorescence microscopy images of MCF-7 and MDA-MB-231 cells stained with DAPI. Chromatin and DNA fragmentation was observed after cells were treated with G4/MTX, G4-siRNA and G4/MTX-siRNA treatment groups that are represented by the arrows.

**Table 1 genes-12-01102-t001:** Sequences of the oligonucleotide primers used for PCR amplification.

Primers	Sequences	
β actin	Forward	5′-TCCCTGGAGAAGAGCTACG-3′
	Reverse	5′-GTAGTTTCGTGGATGCCACA-3′
*HMGA2*	Forward	5′-TGGGAGGAGCGAAATCTAAA-3′
	Reverse	5′-TCCCTGGAGAAGAGCTACG-3′

HMGA2: High mobility group protein A.

## Data Availability

The data that support the findings of this study are available from the corresponding author upon reasonable request.

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
