# Peer review of "Silencing of HMGA2 by siRNA Loaded Methotrexate Functionalized Polyamidoamine Dendrimer for Human Breast Cancer Cell Therapy"

_genes, 2021, doi:10.3390/genes12071102_

Round 1

Reviewer 1 Report

The results of the work are highly relevant and statistical analysis is sound.  However, the methodology employed is highly complex and difficult to interpret by investigators with no particular expertise in biology.

Major comments:

  1. The authors should simplify the M&M and Results sections in order to improve the overall clarity of the manuscript.
  2. The number of abbreviations and figures is excessive and adds complexity to the manuscript.
  3. The authors should include the advantages and limitation paragraph.

Minor comments and suggestions:

  1. M&M, line 166: the authors state that three different cells lines are employed (MCF-7, MDA-MB231, A549), but in the consecutive sections and in the abstract only two cell lines are mentioned (MCF-7, MDA-MB-31). Also in the Introduction (lines 53-54, and 109-110), only these latter types are mentioned as human breast cells. Are the cancer cell lines A549 were not finally used?
  2. Abstract: abbreviations for PAMAM, HMGA2 and siRNA should be provided
  3. M&M, Line 116 and 118: abbreviations for PAMAM and MTX are unnecessarily repeated, respectively.
  4. M&M, Line 174: %70 means 70%?
  5. M&M, Lines 277-278: please explain what it means
  6. Results, Figures: the resolution of the Figure 1 should be improved and all the abbreviations have to be expanded.
  7. Tables: all the abbreviations have to be expanded.
  8. Discussion, line 533: is the word “Intestinally” suitable?

Please find comments and suggestion in the attached file

Reviewer 2 Report

The authors are addressing an important and fundamental question in cancer research, i.e. how to enhance delivery of siRNAs and/or drugs into cancer cells.  However, the interesting and compelling data described here is detracted from by poor presentation.

For example:

  • The quality of English needs to be greatly improved throughout the manuscript. Of note, the introduction does not read smoothly.  For some basic suggested edits please see attached document.
  • All the figures need to be of higher resolution, particularly Figures 1, 6 and 7.
  • The presentation of figures such as Figure 2 needs to be improved.The colours and patterns chosen to present data are confusing and unappealing.  
  • Figure 7- please make the font larger on the histogram quadrants so that percentages of cells in each quadrant can be seen.It is not clear if you are presenting apoptosis rate as annexin-V-fitc positive + PI positive cells.  There could be some further description of this.  e. what percentage of cells are early apoptosis (annexin-V_FITC single positive)?  How many are double positive?
  • Figure 8:I am not sure what this adds to the manuscript without some higher resolution images and quantification.  In your methods it is noted that the cells were washed three times prior to DAPI staining- this is not appropriate in this circumstance, as sick/dying cells will be more loosely attached and will wash off prior to imaging.  Potentially this experiment could be repeated with cells stained with DAPI, then fixed and washed?

Reviewer 3 Report

The work has introduced the G4/MTX:siRNA nano-complex which can be used for binding HMGA2 siRNA. And the release of MTX and siRNA from synthesized complexes is in a time- and pH-dependent manner. By using G4/MTX:HMGA2 siRNA, the study has successfully downregulated HMGA2 in FR overexpressing cancer cells by elevating the apoptosis level in the cells. The author suggested that the newly developed G4/MTX:siRNA nano-complex may be a promising strategy to increase apoptosis caused by HMGA2 suppression, which is the candidate of therapeutic target in human breast cancer.

Overall, the paper has described a novel strategy may be useful for cancer therapy. Due to lacking of strong evidence to show the stable and effective HMGA2 siRNA binding and delivery into the cancer cells via G4 PAMAM/methotrexate (G4/MTX), the manuscript need to offer more clear evidence of success delivery of HMGA2 siRNA via G4/MTX into the cancer cells. It remains unknown that how much G4/MTX bound HMGA2 siRNA uptaken into the cells can be released in the cell. And there is no data to show if G4/MTX bound HMGA2 siRNA is much better than conventional siRNA in knockdown gene expression.

Major concerns:

  1. The author mentioned in the materials and methods that G4/MTX nanosystem was synthesized at ratio 1:16 mol: mol NH2 of G4 dendrimer to COOH of MTX. There is no explains on the reason why to choose the ratio as it. No reference or control tests with different ratios had been used.
  2. Figure legends of Fig.1 are in a big mess. The particle size has not been characterized in the data, either. Please make sure what have been correctly showing in Figure 1C “Atomic force microscopic (AFM) for G4/MTX: siRNA”. By showing it in right order, please clearly show what are the exact nanoparticles in the image. Is there any difference of the size of the nanoparticles before and after binding with HMGA2 siRNA?
  3. It is very doubting that “the release of MTX and siRNA from synthesized complexes was found in a time- and pH-dependent manner”. It is hard to understand what the author would like to express and to show in Figure 1E and 1F. Does it mean that G4/MTX: HMGA2 siRNA is unstable or stable at room temperature? If it is very stable, how HMGA2 siRNA was released from the nanocomplex in the cells after absorption? If it is not, how to optimize the concentration of G4/MTX: HMGA2 siRNA added into the cell cultures? The measurement of the degradation of G4/MTX: HMGA2 nanoparticles after being uptaken into the cells should be more important than what have been shown here.
  4. In 2.8, it says that “The studied cells seeded into the 6-well plats with 2x105 cells and were treated with G4/MTX, G4: siRNA, and G4/MTX: siRNA for 48 h.” Why does it cost that long if the cellular uptake of G4/MTX nanosystem only takes 4 hours as introduced in 2.7.1? How to define the proper concentration for the incubation for such long incubation if the character of HMGB4 siRNA bounded nanoparticle was changed to be different from empty G4/MTX nanosystem?
  5. No data had directly shown siRNA has been delivered and can be release from the G4/MTX complex after uptaken into the cells. It is also tricky because MTX has effects on cancer cells by itself already as the author mentioned in the introduction. Then it is hard to know if the cell apoptosis is caused by HMGA2 inhibition or just because of MTX. If taking a look at Fig.3 and Fig.4 at the same time, it can be easily noticed that FITC and FAM in all those images are the same. The green background of those panel are not at the same level. It is arbitrary data without statistical analyses of fluorescent intensities. And it claims that 4 hours incubation in the figure legends of both figures, which is not identical as described in the methods.
  6. The enhanced apoptosis of nanocomplex is not that strong and it has been described that “HMGA2 silenced MCF-7 and MDA-MB-231 cells showed 23.28% (p values <0.001), 455 and 27.6% (p values <0.001) apoptosis for G4/MTX:siRNA treatment group while G4:siRNA displayed 17.26% (p values <0.001) and 15.94% (p values <0.01) apoptosis respectively in comparison with apoptosis in the control group (Figure 7a and b).” The high apoptotic level of control groups is already abnormal. No control of conventional HMGA2 siRNA transfected cells and MTX treated only controls. These should be added.

Minor concerns:

All figure legends need to be carefully re-written with all required descriptions of the figures. The scale bars need to be added on the cell images.

Round 2

Reviewer 2 Report

The authors have significantly improved the written presentation of the paper, i.e. writing quality.  However, I still have considerable concerns about the figures.  The main concern is that most of the figures are not high enough resolution.  See Fig 1 for prime example.  

Other corrections to be made with figures include, but are not limited to:

  • Improved resolution of western blot in figure 4.  Plus addition of molecular weight markers.
  • Please add a percentage sign "%" after numbers on FACS histogram quadrants.
  • Remove grey background on figures 2 and 3.  
  • Standardize labels on figures.  "A"etc on figure 2/3 is in lower case in a blue circles, but black in no circle in other figures.

There needs to be more attention to detail with these points (and the figures in general) in order for this paper to be at a publishable standard.

Reviewer 3 Report

The authors have answered all my questions and done the intensive revision of the manuscript very carefully. There is no more questions for me on the work. I am pleased to recommend the manuscript to be accept for publication.

Author Response

The authors have answered all my questions and done the intensive revision of the manuscript very carefully. There is no more questions for me on the work. I am pleased to recommend the manuscript to be accept for publication.

Response; Thank you for reading the revised version of manuscript carefully. We were tried to improve the quality of manuscript based on your valuble comments .